# Electronic Structure of NdFeCoB Oxide Magnetic Particles Studied by DFT Calculations and XPS

**DOI:** 10.3390/ma16031154

**Published:** 2023-01-29

**Authors:** Vadim Yu. Samardak, Alexander A. Komissarov, Alexander A. Dotsenko, Vladimir V. Korochentsev, Ivan S. Osmushko, Anton A. Belov, Pavel S. Mushtuk, Valerii A. Antonov, Ghader Ahmadpour, Farzad Nasirpouri, Alexander S. Samardak, Alexey V. Ognev

**Affiliations:** 1Laboratory of Spin-Orbitronics, Institute of High Technologies and Advanced Materials, Far Eastern Federal University, 10 Ajax Bay, Russky Island, Vladivostok 690922, Russia; 2Laboratory of Electronic Structure and Quantum Chemical Modeling, Institute of High Technologies and Advanced Materials, Far Eastern Federal University, 10 Ajax Bay, Russky Island, Vladivostok 690922, Russia; 3Faculty of Materials Engineering, Sahand University of Technology, Tabriz 5513351996, Iran

**Keywords:** NdFeCoB oxide magnetic particle, electronic structures, photoelectron spectroscopy, quantum chemical modeling

## Abstract

Neodymium-iron-boron magnetic oxide powders synthesized by sol–gel Pechini method were studied by using X-ray photoelectron spectroscopy (XPS) and quantum chemical modeling. The powder structure was examined by using X-ray diffraction (XRD) and modeled by using density functional theory (DFT) approximation. The electronic structures of the core and valent regions were determined experimentally by using X-ray photoelectron spectroscopy and modeled by using quantum chemical methods. This study provides important insights into the electronic structure and chemical bonding of atoms of NdFeCoB oxide particles with the partial substitution of Fe by Co atoms.

## 1. Introduction

The most permanent magnets are neodymium-iron-boron (Nd_2_Fe_14_B) magnets, which are widely used as components of electric motors in electric and hybrid cars, in wind turbines, as well as in computer hard drives and other electronic devices [1,2]. The use of neodymium magnets that create a strong magnetic field is promising in many areas. In addition to bulk magnets, magnetic powders are also important for practical use. Magnetic powders with particle sizes ranging from nanometers to micrometers are used in sorbents, magnetic liquids, magnetic elastomers, and polymer magnets [3,4]. Magnetic powders make it possible to create composites exhibiting the theoretically predicted maximum energy ratio of (BH)_max_ [5]. Physical methods are used to produce powders, for example, by ball milling with or without surfactant, namely the mechanosynthesis method [6,7,8,9]. Furthermore, with the aid of chemical synthesis, it is possible to implement “bottom-up” approaches. For instance, magnetic nanopowders are obtained first in oxide forms and then in reduced forms using the sol–gel method, microwave processing, and thermal decomposition [10,11,12]. The possibility of introducing additives during the synthesis process improves the characteristics of magnetically hard nanomaterials and reduces the cost of their manufacture. For example, tailoring the composition of Nd-Fe-B provides magnets with more efficient properties. The effect of the substitution of iron for cobalt on the magnetic properties and microstructure of Nd-Fe and Co-B particles obtained by using the Pechini sol–gel method has been discussed in our recent paper [10].

In the synthesis of powders, it is crucial to control the phase composition and electronic structure at each stage, because the hard magnetic properties of the resulting material depend on this. In this paper, we showed how the electronic structure upon substitution of Fe by Co in oxidized samples of NdFe_(1−x)_Co_x_B can be investigated using XPS spectroscopy. The use of quantum-chemical calculations allowed us to resolve the structure of the detected phases. The concentrations of cobalt, the qualitative composition of the samples, and the effect of the substitution of iron for cobalt on the electronic structure and X-ray spectra of the samples were determined for the oxidized samples after synthesis.

## 2. Materials and Methods

### 2.1. Experiment

Three samples containing the oxidized phases of NdFe_(1−x)_Co_x_B were investigated: a sample without Co (x = 0), a sample with an ultralow concentration of Co (x = 0.05), and a sample with an equal content of Fe and Co (x = 0.5). The synthesis method was described in detail in our previous work. The results of the complex studies of the structure and magnetic properties obtained by using X-ray diffraction, scanning and transmission electron microscopy, energy dispersive X-ray spectroscopy (EDX), X-ray photoelectron spectroscopy (XPS), vibrating ample magnetometry, and first-order reversal curve methods (FORC diagram) were extensively presented [10].

In this research, the experimental information about internal and valence electrons was obtained using the method of XPS. XPS spectra were measured on an Omicron ultrahigh-vacuum photoelectron spectrometer with a hemispherical electrostatic analyzer (radius of curvature of 125 mm) and an AlKα radiation source (source energy of 1486.7 eV). The spectra were processed using the CASA XPS program Version 2.3.12 [13]. The electron binding energy scale was calibrated using the method of the internal standard according to the binding energy of C1s (285.0 eV). The survey spectra were obtained with a pass energy of 50 eV and some parts of the spectra were obtained with a pass energy of 20 eV. The atomic concentrations were estimated from the intensities of the XPS spectra of internal levels.

Phases were determined using the Bruker X-ray diffractometer D8 (CuKα radiation source) and the Rietveld method in the MAUD program Version 2.992 (by Luca Lutterotti).

### 2.2. Calculations

#### 2.2.1. Methods

To interpret the photoelectron spectra and describe the electronic structure of the samples under study, quantum chemical modeling was carried out using the density functional theory (DFT) method. To calculate the level energy of the molecular models, we chose the popular functional B3LYP [14] and the basis set Def2-SVP [15], which includes the 4f lanthanide subshells. The quantum chemical modeling of molecules was performed using the GAMESS software package [16].

The electronic structure of the crystals was calculated in the DFT approximation using the Quantum Espresso software package [17]. The simulation of the density of states of the crystal was performed using the Vanderbilt USSP pseudopotential. The exchange-correlation effects were considered using the PBE functional. Integration over the Brillouin zone was carried out using a grid of k-points: 3 × 3 × 3 for Co_3_O_4_, 2 × 2 × 2 for Fe_2_O_3_ and NdFeO_3_, 2 × 2 × 1 for NdBO_3_, and 1 × 1 × 1 for Nd_2_Fe_14_B with the conventional Gaussian distribution (ordinary Gaussian spreading). The convergence criterion was 10^−6^.

#### 2.2.2. Molecular and Crystal Models

Different molecular models were created for quantum chemical modeling by using the DFT method (see Table 1). The models were obtained on the basis of calculated crystal structures (initial crystal structures were taken from the electronic database of the Materials Project [18]). The molecular orbital (MO) energies of the Co_3_O_4_, NdFeO_3_, CoNdFeO_4_, and Fe models were calculated, and the optimal geometry and the MO energies of the NdBO_3_ models were calculated.

## 3. Results

### 3.1. Estimation of Concentrations According to XRD Data

From the analysis of the XRD data (Appendix A), the NdFe_(1−x)_Co_x_B sample with x = 0 contains three compounds: Fe_2_O_3_, NdBO_3_, and NdFeO_3_. The sample with x = 0.05 contains three compounds Fe_2_O_3_, Fe_3_O_4_, and CoNdFeO_4_. In the sample with x = 0.5, there are four compounds: Fe_3_O_4_, NdFeO_3_, Co_3_O_4_, and NdBO_3_. The relative composition according to the XRD data is given in Table 2.

### 3.2. Estimation of Concentrations According to XPS Data

The X-ray photoelectron spectra of the samples with applied positions and areas of the main bands and atomic concentrations are given in the Appendix A. The atomic concentrations according to the XPS data are given in Table 3.

All survey spectra show an intense band around 1071 eV on the binding energy scale. This band is presumably assigned to Na1s. The presence of sodium in the samples may be associated with the synthesis of hydrochloric acid, which is necessary for the creation of metal chlorides in the process of synthesizing the samples.

Boron is not detected in the survey spectra; however, the low photoionization cross section of boron allows its insignificant concentration with the intensity of the B1s band at the noise level. Nitrogen is also not detected in the spectra.

According to the established concentrations, the approximate ratio of the compounds was determined in the samples (Table 4).

### 3.3. Spectra Modeling

This section presents the experimental and theoretical spectra of the NdFe_(1−x)_Co_x_B samples with x = 0, x = 0.05, and x = 0.5. The intensities of the theoretical spectra were obtained from the concentrations based on the XPS spectra and the XRD data. The height of the “stick” spectrum lines is proportional to the experimental photoionization cross section at a photon energy of 1486.7 eV, the Mulliken contribution of the shell to the corresponding molecular orbital, the number of atoms, and the level degeneracy for the selected binding energy scale. The theoretical spectra of the O1s levels are the sum of Gaussians plotted on individual MOs with a half-width equal to the analyzer’s bandwidth (approximately 1.5 eV).

#### 3.3.1. Oxide Sample with Co Concentration x = 0

In Figure 1, a theoretical “spectrum” is plotted under the experimental survey spectrum. The intensity ratio for the XPS data “spectrum” lines is chosen in accordance with the concentrations—1:2:2 for Fe_6_O_9_:NdBO_3_:NdFeO_3_. The MO energies of the three model compounds are plotted under the theoretical “spectrum”. The overview spectrum is satisfactorily described by the concentrations according to the XPS and the XRD.

Figure 2a shows the experimental XPS spectrum of the valence region (up to 30 eV on the binding energy scale). The 19–24 eV band (in the binding energy scale, in the figure above) is dominated by the Nd5p and O2s levels. The largest contribution to the 4–10 eV band comes from Nd4f and Fe3d. The calculation method used shifts the energy of Nd4f by 3–4 eV toward higher binding energies (to the left in the spectrum).

Figure 2b shows the spectrum of the O1s level. Neither the DFT nor the ab initio methods give an absolute correspondence between the calculated MO energies and the experimental data. However, the change in the level energy depending on the environment of the atom is transmitted quite accurately.

Despite the primitiveness of the chosen models for the compounds, the composition of the experimental O1s band can be estimated from the calculations. The molecular orbital energy of the oxygen that is bound to iron (Fe_6_O_9_ model) is the lowest, which means the highest binding energy. Oxygen bound to neodymium and iron provides less binding energy for O1s electrons (NdFeO_3_ model), and oxygen bound to boron provides the lowest binding energy (NdBO_3_ model).

Since the calculation does not describe the spin–orbit interaction, the splitting of the Nd3d_5/2_ and Nd3d_3/2_ levels for the theoretical spectrum is taken from the XPS spectrum and equal to 22.4 eV (see Figure 3). The peak for Nd3d_5/2_ is found at a binding energy of 981.8 eV, and the peak for Nd3d_3/2_ is at an energy of 1004.2 eV.

#### 3.3.2. Oxide Sample with Co Concentration x = 0.05

The X-ray photoelectron spectra of the oxide sample with x = 0.05 are shown in Figure 4 and Figure 5. The theoretical spectral intensities for the survey spectrum, the valence region spectrum, and the O1s spectrum were calculated based on the concentrations obtained by the two methods, XPS and XRD.

On the overview spectrum (Figure 4), the relative intensities of Nd3d, compared to the experiment, are better transmitted by the XRD concentrations (the spin orbital splitting of theoretical levels is taken from the XPS spectrum and is equal to 22.4 eV). The Co2p intensities are more closely reproduced by the XPS concentrations, from which it can be concluded that the cobalt content found in the sample in the XPS spectrum is x ≈ 0.25.

It can be seen from the theoretical survey spectrum that, under the condition x = 0.05, the Fe2p and Co2p bands should differ significantly in intensity (by a factor of 4–5), which is not observed in the experimental XPS spectrum. The C1s band, as in the case of the sample without cobalt, is due to organic dirt in the analyzer chamber.

Figure 5a shows the spectrum of the valence region. The Nd5p and O2s levels contribute to the band in the 18–25 eV region. The levels of Nd4f, Co3d, and Fe3d contribute to the 4–8 eV band. A stick spectrum based on the XRD concentrations better conveys the structure of the valence region.

The shape of the O1s band is better represented by the theoretical spectrum based on the XPS concentrations (Figure 5b). According to the calculations, the binding energy of electrons in the O1s subshell chemically bound to neodymium is 2.2 to 2.5 eV lower than that of 3d metal oxides.

#### 3.3.3. Oxide Sample with Co Concentration x = 0.5

The X-ray photoelectron spectra of an unreduced sample with x = 0.5 are shown in Figure 6 and Figure 7. The intensities used in the modeling of the survey spectrum, the O1s spectrum, and the valence region spectrum were calculated based on the concentrations obtained by the two methods, XPS and XRD.

The experimental survey spectrum (Figure 6) shows a high oxygen content compared to the theoretical spectra. This indicates a high content of impurity in (dirty) oxygen along with carbon. The relative intensities of Co2p and Fe2p in the theoretical spectrum, based on the XPS concentration, better describe the experiment and correspond to the cobalt content x = 0.5.

Three bands are visible in the valence spectrum (Figure 7a). The two intense broad bands in the binding energy ranges of 18–23 eV and 3–7 eV are due to ionization from the Nd5p, O2s and Nd4f, Co3d, and Fe3d levels, respectively. The contribution of boron is insignificant as a result of its low photoionization cross section. Impurity in carbon (levels C2s and C2p) contributes insignificantly (but noticeably in the region of 11–14 eV) to the background in the range of 5–20 eV.

A small band in the 30–31 eV region can be attributed to the Na2p level, which has a high relative photoionization cross section and a concentration of approximately 2.4% (Table 3). The band in the region of −34 eV of molecular orbital energy depicted in the theoretical spectrum belongs to Nd5s. The calculation gave an erroneous value for the energy of this level (about −34 eV of the MO energy), and the Nd5s shell does not contribute to the band at 30–31 eV on the binding energy scale.

In the O1s spectrum (Figure 7b), a low-intensity part of the band in the region of 534–536 eV of the binding energy (scale is located at the top of the image) can be attributed to oxygen atoms being chemically bonded to boron. Oxygens bound with iron and cobalt in the Fe3O4 and Co3O4 oxides, respectively, and contribute to the region of 531–532 eV. The intense part of the band in the region of lower binding energies (529–530 eV) is occupied by neodymium-bound oxygens.

### 3.4. Modeling Spectra from Crystal Calculations

The theoretical photoelectron spectra were constructed by expansion in terms of the Gaussian function, considering the instrumental function of 1.5 eV and the data on the photoionization cross sections for the AO contributions obtained in MO for the LCAO method and spd functions for the PAW method (Figure 8).

The calculation of contributions in the LCAO method was performed using MPA (Mulliken Population Analysis) [19] using the expansion of MO in linear combinations of AO. In this method, the electron density of an atom is determined by the sum of the squares of the expansion coefficients of the MO over the AO of a given atom plus half the overlap density (overlap population) of the orbitals of a given atom with a neighboring atom. The height of the function corresponding to the intensity in the spectrum is determined as the sum of the product of each contribution of the atomic orbital to the MO multiplied by the corresponding ionization cross section of the electron shell of a free atom for a radiation energy of 1486.6 eV (AlKa).

The full theoretical spectrum is the sum of Gaussian functions. Each function corresponds to a separate state.

The position of the maxima of the Gaussian function on the energy scale corresponds to the energy of the state. The ionization cross sections have the following values: B2s = 28, B2p = 24, O2s = 190, O2p = 24, Co4s = 71, Co3p = 2600, Co3d = 370, Fe4s = 70, Fe3p = 2200, Fe3d = 220, Nd6s = 34, Nd5p = 970, and Nd4d = 11,030, in relative units. The half-width at half-height is set equal to the instrumental function of the spectrometer and is 1.5 eV.

## 4. Conclusions

In this work, the electronic structure of NdFe_(1−x)_Co_x_B oxide powders with different cobalt content is shown using XPS and DFT methods. Elemental analysis of the XPS method gave a different qualitative composition of the samples from that of the XRD method; both concentrations (according to the XPS and XRD) were used to model the spectra. The modeling the spectra of the molecular models by using quantum chemical methods made it possible to qualitatively describe the structure of the core levels of the X-ray spectra.

A comparison of the results of calculations based on the molecular orbital method with the spectra makes it possible to determine how the atoms in the samples are chemically bound. This approach is well applicable to the core levels, but poorly describes the valence region of the spectrum. For modeling the spectra of the valence region of solids, plane-wave methods are more suitable for calculating the electronic structure of a crystal structure.

Using a combination of the experimental method (XPS) and calculations will make it possible to investigate changes in the electronic structure and better understand the reasons for changes in thermostability when Nd base magnets are doped with Co or other additives.

## Figures and Tables

**Figure 1 materials-16-01154-f001:**
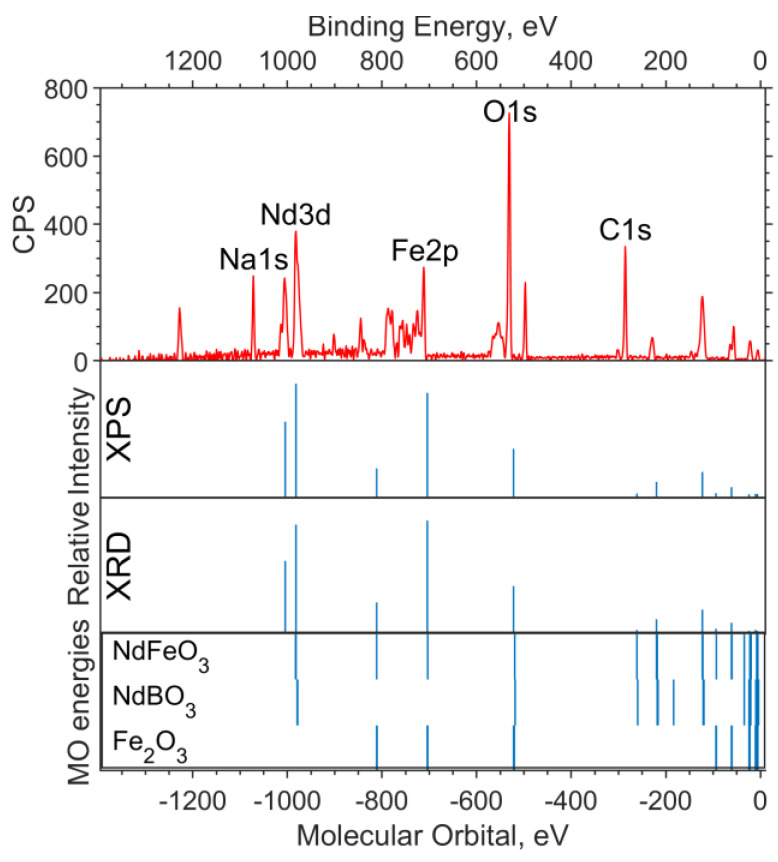
Overview of the XPS spectrum of the NdFe_(1−x)_Co_x_B oxide powders with x = 0, the theoretical “stick” “spectrum”, and the MO energies of the model compounds.

**Figure 2 materials-16-01154-f002:**
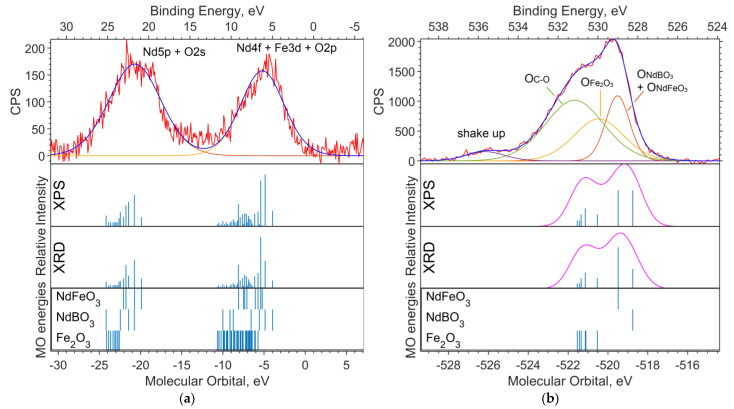
(**a**) XPS spectrum of the valence region and (**b**) spectrum of oxygen 1s levels for the NdFe_(1−x)_Co_x_B oxide powders with x = 0. The Gaussian functions with a half-width FWMH = 1.5 eV are superimposed on the theoretical intensities on (**b**).

**Figure 3 materials-16-01154-f003:**
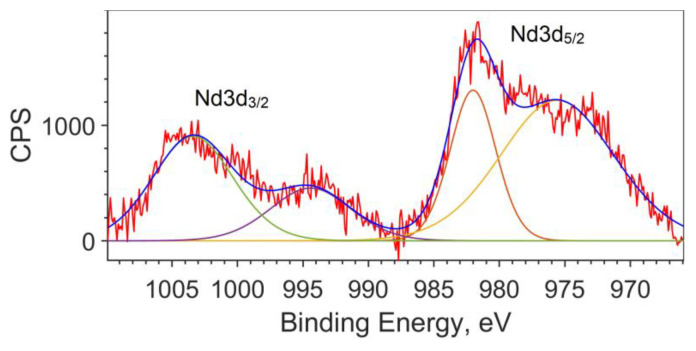
Spin–orbit splitting of Nd3d for the NdFe_(1−x)_Co_x_B oxide powders with x = 0.

**Figure 4 materials-16-01154-f004:**
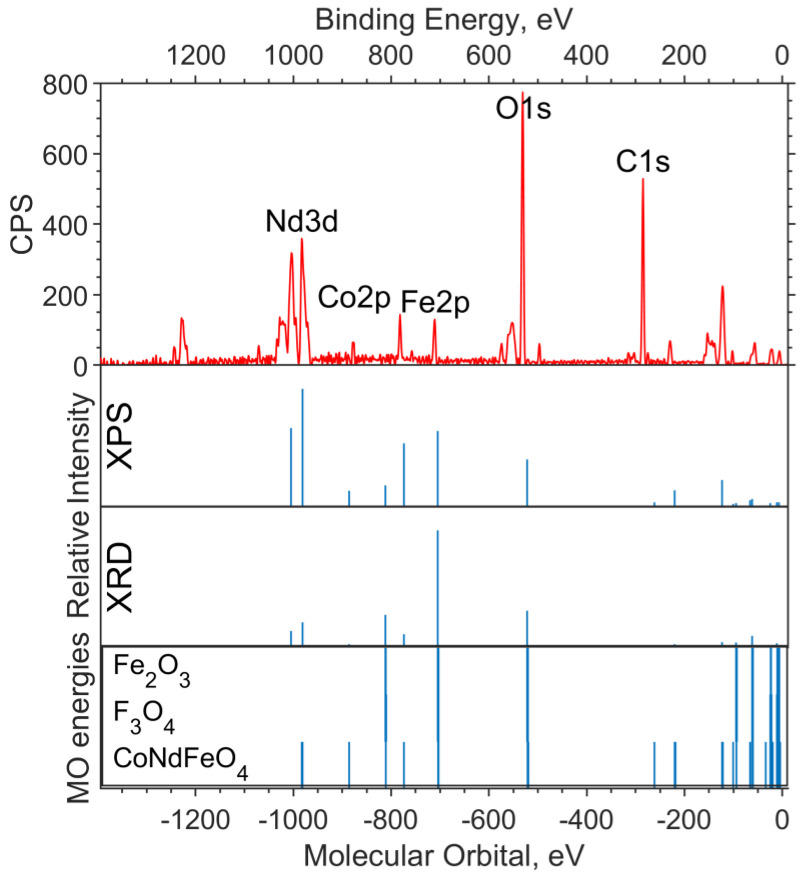
Survey XPS spectrum for the NdFe_(1−x)_Co_x_B oxide powders with x = 0.05 (the background is subtracted, and smoothing is applied), the theoretical “stick” “spectrum” (intensities by the two concentrations are presented: on top—intensities based on XPS concentrations, bottom—intensities based on XRD concentrations), and the MO energies of the model compounds.

**Figure 5 materials-16-01154-f005:**
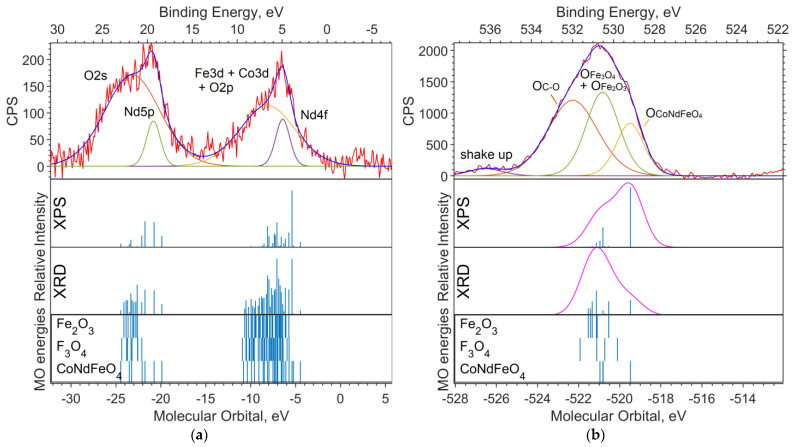
(**a**) XPS spectrum of the valence region and (**b**) spectrum of oxygen 1s levels for the NdFe_(1−x)_Co_x_B oxide powders with x = 0.05. The theoretical spectral intensities based on XPS (top) and XRD (bottom) concentrations, and the MO energies of the model compounds with Gaussian functions with a half-width FWMH = 1.5 eV are superimposed on the theoretical intensities on (**b**).

**Figure 6 materials-16-01154-f006:**
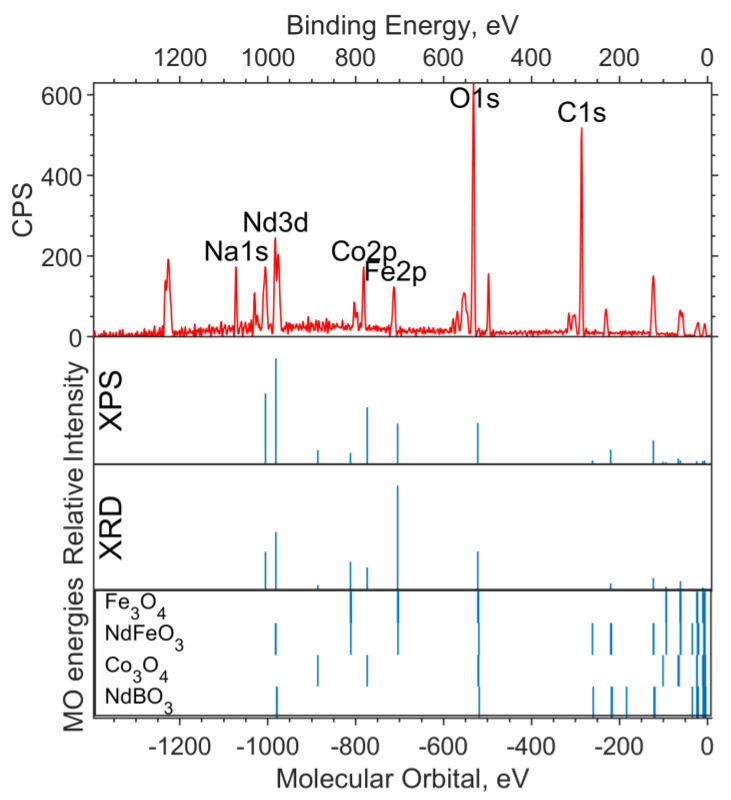
Survey XPS spectrum for the NdFe_(1−x)_Co_x_B oxide powders with x = 0.5 (the background is subtracted, and smoothing is applied), the theoretical “stick” “spectrum” (intensities by the two concentrations are presented: on top—intensities based on XPS concentrations, bottom—intensities based on XRD concentrations), and the MO energies of the model compounds.

**Figure 7 materials-16-01154-f007:**
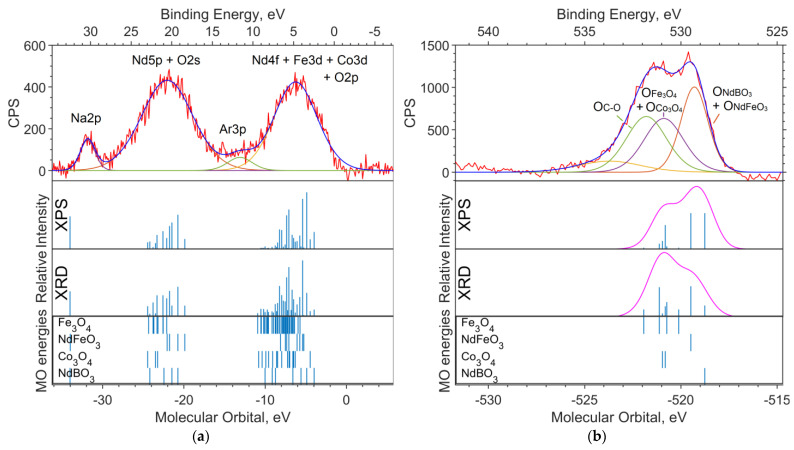
(**a**) XPS spectrum of the valence region and (**b**) spectrum of oxygen 1s levels for the NdFe_(1−x)_Co_x_B oxide powders with x = 0.5. The theoretical spectrum intensities based on the XPS (top) and the XRD (bottom) concentrations, and the MO energies of model compounds with Gaussian functions with a half-width FWMH = 1.5 eV are superimposed on the theoretical intensities on (**b**).

**Figure 8 materials-16-01154-f008:**
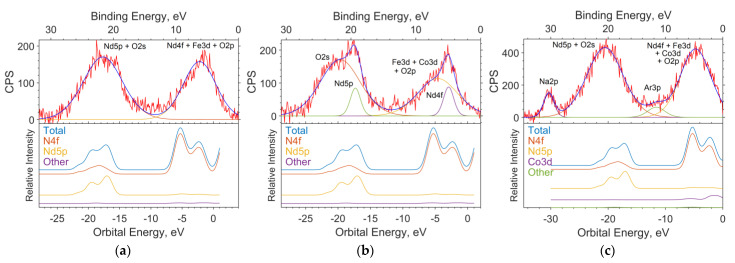
The spectrums of the valence region of the samples with x = 0 (**a**), 0.05 (**b**), and 0.5 (**c**); the theoretical spectrums based on the results of solid-state simulation; and the contribution of individual subshells to the spectrum.

**Table 1 materials-16-01154-t001:** Molecular and crystal models of compounds.

Compound	Molecular Model	Crystal Model
Fe_2_O_3_ (molecular model—Fe_6_O_9_)	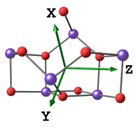	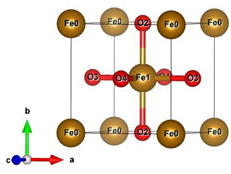
NdBO_3_	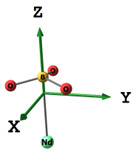	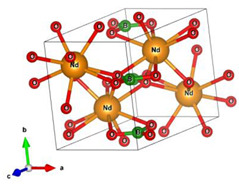
NdFeO_3_	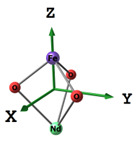	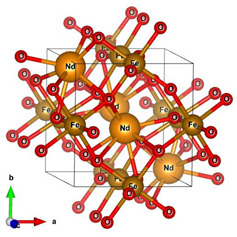
Co_3_O_4_	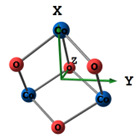	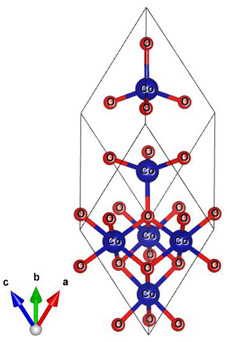
CoNdFeO_4_	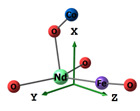	-
Nd_2_Fe_14_B	-	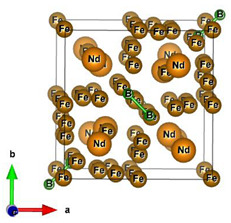

**Table 2 materials-16-01154-t002:** Composition of NdFe_(1−x)_Co_x_B according to the XRD data.

x	Phase	Weight (%)
0	NdFeO_3_	37.9 (3)
	NdBO_3_	18.0 (2)
	Fe_2_O_3_	44.1 (3)
0.05	Fe_2_O_3_	72.0 (7)
	Fe_3_O_4_	9.8 (8)
	CoNdFeO_4_	18.0 (4)
0.5	NdFeO_3_	25.7 (3)
	Fe_3_O_4_	55.8 (4)
	NdBO_3_	9.6 (2)
	Co_3_O_4_	8.9 (5)

**Table 3 materials-16-01154-t003:** Concentrations of identified atoms from the NdFe_(1−x)_Co_x_B oxide samples.

Atomic Shell	Pos.	Area	at.%
**x = 0**			
Nd3d_5/2_	982	4117.0	6.3
Fe2p_3/2_	710	1705.0	12.5
O1s	531	3807.5	36.6
C1s	285	1460.0	41.1
Na1s	1071	1054.0	3.5
**x = 0.05**			
Nd3d_5/2_	982	5161.0	3.41
Fe2p_3/2_	710	1260.5	2.64
C1s	285	2551.0	56.14
O1s	531	4340.0	32.60
Co2p_3/2_	781	2343.7	2.78
Na1s	1071	939.0	2.43
**x = 0.5**			
Nd3d_5/2_	982	8374.5	3.15
Co2p_3/2_	781	2737.0	3.02
Fe2p_3/2_	710	1462.0	2.95
C1s	285	3028.0	64.03
O1s	531	3386.5	24.44
Na1s	1071	972.0	2.41

Pos. is the position on the spectrum according to the binding energy, eV; Area is the height area; arbitrary units; at.% is the concentration of the shell corresponding atoms.

**Table 4 materials-16-01154-t004:** Approximate relative concentrations of the NdFe_(1−x)_Co_x_B oxide compounds on the basis of analysis of atomic concentrations from X-ray photoelectron spectra. The given concentrations are used to construct the model spectra.

**x = 0**	
NdFeO_3_	2
NdBO_3_	2
Fe_2_O_3_	3
**x = 0.05**	
Fe_2_O_3_	1
Fe_3_O_4_	1
CoNdFeO_4_	10
**x = 0.5**	
Fe_3_O_4_	1
NdFeO_3_	3
Co_3_O_4_	2
NdBO_3_	3

## Data Availability

Not applicable.

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
