# Peer review of "Electronic Structure of NdFeCoB Oxide Magnetic Particles Studied by DFT Calculations and XPS"

_materials, 2023, doi:10.3390/ma16031154_

Round 1

Reviewer 1 Report

This manuscript employed XPS, XRD, and DFT to study the electronic structure of NdFeCoB oxide fabricated by the sol-gel Pechini method. This work reported the change in the electronic structure of NdFeCoB oxide with the Co concentration. However, to improve the current manuscript, I believe the following comments should be addressed:

Comments 1): It is extremely important to properly calibrate the binding energy to correctly analyze the XPS spectra. The ASTM E2108-16 standard (DOI: 10.1520/E2108-16) requires the use of Au, Ag, and Cu foils to calibrate the spectrometer of the analyzer. The authors used C 1s at 285 eV as the reference for binding energy calibration. This is highly unreliable. See Greczynski G., Hultman L., X-ray photoelectron spectroscopy: Towards reliable binding energy referencing., Prog Mater Sci 107, 100591 (2020). The binding energy has to be carefully calibrated to draw any conclusion from the XPS spectra in this work.

Comments 2): The Materials and Methods section needs to be carefully revised. In line 66, the reference to CASAXPS is missing. In line 68-69, the author mentioned: “analyzer bandwidth”. Please elaborate on the meaning of “analyzer bandwidth” and its difference to pass energy.

In the 2.1.2 XRD section, the authors include experimental results in the methods section. Please move this to the corresponding results section.

Comments 3): An unknown peak is detected at ~ 500 eV in the XPS spectra. This peak is not from any elements reported in this work. Please assign the peak core level and its element. Please discuss the impact of this impurity on the electronic properties.

Comments 4): Boron was not detected in all the XPS survey scans due to the low sensitivity factor. Is B detectable at high-resolution scans (0.05 eV steps)? The existence of B is critical for the correctness of the conclusions in this work.

Comments 5): The baselines and sum lines are not shown in all the fitted XPS spectra. The meaning of the fitted peaks is also missing. This casts doubt on the accuracy of the XPS analysis. Please revise accordingly.

Comments 6): In line 186, the author attributes the carbon signal to the “organic dirt” in the UHV. This is not reasonable. Compared to the carbon contamination from the air, the carbon from UHV is trivial.

Comments 7): Academic writing has to be precise. In line 121, The author mentions “ X-ray electron spectra.” What is its difference from “X-ray photoelectron spectra”?

In lines 33, 59, and 154, the full names of abbreviations are missing.

References in line 26-29 need to be added.

Reviewer 2 Report

In this work, neodymium-iron-boron magnetic oxide powders synthesized by sol-gel Pechini method were studied by X-ray photoelectron spectroscopy (XPS) and quantum chemical modeling. But there are still many issues to be resolved. Major revision are recommended.

1.     The author obtained the electronic structure of NdFeCoB only through three components x=0,0.05 and 0.5, which lacked data support.

2.     The author uses XRD to analyze samples with different Co doping amounts, but the XRD data in the supporting information is difficult to get data, so it is suggested to reassemble the Fig. S1.

3.     The author refined the XRD data using the Rietveld method, then the parameters of the refining process, such as R factor, COF, etc., should be given. Without this information, refining data is hard to convince.

4.     How did the author test the XRD data? It can be seen from Fig.S1 and S2 that the received X-ray signal is very weak.

5.     In the process of XPS data analysis, the author mentioned the use of data smoothing and contamination in the sample cavity, including the initial use of XRD data for the initial model calculation. Are the results obtained through so many errors trustworthy?

Reviewer 3 Report

In the present work the authors have studied and presented the electronic structure of NdFe(1-x)Cox oxide powders by varying the concentration of cobalt (x) by employing the XPS and DFT methods. The topic of the research work is interesting. However, before I recommend this work for publication in “Materials”, I would like to make the following suggestion. 

 The descriptions of the results and the discussions thereof are too short to develop a clear understanding of the electronic structure of NdFe(1-x)Coxoxide powders. The manuscript requires a major revision in this regard. 

 Also, I would like to raise the following points before the authors.

 (1) Is PBE functional adequate to describe the electronic structure of Nd and transition metals (Fe and Co)?  Have the authors cross-checked the electronic structures of the system under question by employing the density functional theory with Hubbard-type on-site Coulomb potential ( U ) to treat the localized 4f and 3d electrons of Nd and transition metals, respectively?  

 (2) In the “Introduction” the authors have stated that, “tailoring the composition of Nd-Fe-B provides magnets with more efficient properties”. Which composition are they referring to? Is it Nd2Fe14B?

(3) Though the systems are magnetic, a clear description of the magnetic states is missing.

(4) Do the transition metals exhibit ordering? If so, how does the phenomenon affect the properties of the system?

(5) It is not very clear as to what the authors intend to achieve by the process of Co substitution. They need to elaborate upon this. 

(6) What is CPS?

Round 2

Reviewer 1 Report

The manuscript has been improved significantly. I suggest accepting this version for publication.

Author Response

Thank you for your replies 

Reviewer 2 Report

This is a good paper for publication in this journal

Author Response

Thank you for your replies 

Reviewer 3 Report

In the revised manuscript and the response, the points raised in the previous review report are not well and adequately addressed by the authors. I, therefore, cannot recommend this work for publication.

Author Response

Dear reviewer

We addressed all your questions, but we can't adequately  answer to your last comment. Taking into account this situation, we would like to ask the editor to make a decision based on two positive reviews, or send the manuscript and reviews for an additional review.